# Non-spherical microparticle shape in Antarctica during the last glacial period affects dust volume-related metrics

Aaron Chesler[1,2,*], Dominic Winski[1,2], Karl Kreutz[1,2], Bess Koffman[3], Erich Osterberg[4], David Ferris[4], Zayta Thundercloud[4], Joseph Mohan[1,5], Jihong Cole-Dai[6], Mark Wells[7], Michael Handley[1], Aaron Putnam[2], Katherine Anderson[4], and Natalie Harmon[2]

[1]Climate Change Institute, University of Maine, Orono, Maine, 04469, USA
[2] School of Earth and Climate Sciences, University of Maine, Orono, Maine, 04469, USA
[3]Department of Geology, Colby College, Waterville, Maine, 04903, USA
[4]Department of Earth Science, Dartmouth College, Hanover, New Hampshire, 03755, USA
[5]St. Croix Watershed Research Station, Science Museum of Minnesota, Marine on St. Croix, Minnesota, 55047
[6]Department of Chemistry and Biochemistry, South Dakota State University, Brookings, South Dakota, 57007
[7]School of Marine Sciences, University of Maine, Orono, Maine, 04469, USA
*now at: Environmental Studies Program, Goucher College, Towson, Maryland, 21204, USA

*Correspondence to*: Aaron Chesler (aaron.chesler@maine.edu)

**Abstract.** Knowledge of microparticle geometry is essential for accurate calculation of ice core volume-related dust metrics (mass, flux, and particle size distributions) and subsequent paleoclimate interpretations, yet particle shape data remain sparse in Antarctica. Here we present 41 discrete particle shape measurements, volume calculations, and calibrated continuous particle timeseries spanning 50 – 10 ka from the South Pole Ice Core (SPC14) to assess particle shape characteristics and variability. We used FlowCAM, a dynamic particle imaging instrument, to measure aspect ratios (width divided by length) of microparticles. We then compared those results to Coulter Counter measurements on the same set of samples as well as high-resolution laser-based (Abakus) data collected from the SPC14 core during continuous flow analysis. The 41 discrete samples (temporal resolution of ~490 years per sample in the Last Glacial Maximum; LGM) were collected during three periods of millennial scale variability: Heinrich Stadial 1 (18 – 16 ka, n = 6), the LGM (27 – 18 ka, n = 19), and during both Heinrich Stadial 4 (42 - 36 ka, n = 8) and Heinrich Stadial 5 (50 – 46 ka, n = 8). Using FlowCAM measurements, we calculated different particle size distributions (PSDs) for spherical and ellipsoidal volume estimates. Our calculated volumes were then compared to published Abakus calibration techniques. We found that Abakus-derived PSDs calculated assuming ellipsoidal, rather than spherical, particle shapes provide a more accurate representation of PSDs measured by Coulter Counter, reducing Abakus-to-Coulter Counter flux and mass ratios from 1.82 (spherical assumption) to 0.79 and 1.20 (ellipsoidal assumptions; 1 being a perfect match). Coarser particles (>5.0 µm diameter) show greater variation in measured aspect ratios than finer particles (<5.0 µm). While fine particle volumes can be accurately estimated using the spherical assumption, applying the same assumption to coarse particles has a large effect on inferred particle volumes. Temporally, coarse and fine particle aspect ratios do not significantly change within or among the three time periods (p-value > 0.05), suggesting that long range transport of dust is likely dominated by clay minerals and other elongated minerals.

**Copyright statement**

**1 Introduction**

Ice core microparticle (i.e., insoluble dust) data provide a critical perspective on past climate variability because a range of dust metrics (e.g., number and mass concentrations, flux, volume, size distribution, and geochemical composition) can be measured and used to reconstruct atmospheric circulation (i.e., Delmonte et al., 2002; Koffman et al., 2014; Aarons et al., 2017; Wegner et al., 2015; Lambert et al., 2012; Petit et al., 1999), radiation balance (i.e., Lambert et al., 2013; Durant et al., 2009; Baggenstos et al., 2019), and chemical delivery to ocean and terrestrial ecosystems (e.g., Conway et al., 2015; Edwards et al., 2006; Wolff et al., 2006; Spolaor et al., 2013). There are two analytical techniques commonly used to measure dust particle size and concentration in ice cores: Coulter Counter (CC) and continuous Abakus laser particle sensor (Abakus). Each of these methods involve assumptions regarding particle shape. The CC measures particle concentration through electrical resistance in discrete samples, where a solution is fed through an aperture which disrupts a constant electrical field between two electrodes. The impedance of the electrical signal is directly proportional to the particle volume,

and therefore CC data are generally assumed to be the most accurate estimate of particle volume. The Abakus measures particles via disruption of transmitted light; as particles flow through the sensor, the interruption of the light source produces scattering and shadowing resulting in a negative peak in transmitted light, which is used to measure the diameter of the

particle (Ruth et al., 2002). A main advantage of the Abakus is that it can be used in a continuous flow analysis (CFA) system, providing theoretical millimeter-scale resolution, and allowing for in-line comparison with other CFA ice core data (e.g., soluble ions, black carbon, stable isotopes, electrical conductivity). However, it only provides a particle geometric size measurements in one dimension (i.e., particle length or width), and does not provide depth/height information for individual particles (Ruth et al., 2002; Simonsen et al., 2018). Abakus data have traditionally been interpreted using the assumption that

the measured length of each particle is equal to the diameter of that particle and that all microparticles are spherical. While both approaches have advantages, neither provides direct information about particle shape.

Recent work has tested assumptions about particle sphericity by using a third technique, the Single Particle Extinction and Scattering (SPES) method, and found that the spherical approximation leads to mismatches not only in volume but also in calculated particle size distribution (PSD) between Abakus and CC data (Simonsen et al., 2018; Villa et

al., 2016; Potenza et al., 2016; Potenza et al., 2015). SPES uses particle light scattering and absorption measurements to identify extinction cross-sections. Particles are directed through the focal point of a light beam while sensors in the far field collect the total power removed from light scattering and absorption of the particle. Because the sum of scattering and absorption by each particle is proportional to the extinction cross section, particle shape information can be identified for particles relatively quickly and efficiently (Potenza et al., 2015). Using a combination of SPES, CC, and Abakus methods,

Simonsen et al. (2018) provided the first calibration scheme to correct for offsets in the PSD of Abakus-derived data based on particle shape for the Holocene and Last Glacial Period in Greenland.

We expand on previous work by developing and presenting here new dust data from the South Pole ice core (SPC14) that spans 50 – 10 ka and includes Abakus, CC, and particle shape measurements. We identified three intervals of interest (Heinrich Stadial 1 [HS1; 18 – 16 ka], the Last Glacial Maximum [LGM; 27 – 18 ka], and Heinrich Stadial 4 and

Heinrich Stadial 5 [HS4; 42 – 36 ka; 50 – 46 ka, respectively]) based on periods of millennial scale climate changes in stable isotope, microparticle concentration, and $CO_2$ data (Figure 1). SPC14 has a relatively high accumulation rate (7.4 cm/yr during the Holocene and about 3.7 cm/yr during the LGM) and thus a higher temporal resolution compared with many other sites in East Antarctica (Winski et al., 2019; Casey et al., 2014; Lazzara et al., 2012; Winski et al., 2021; Kahle et al., 2020). To assess the size- and time-dependent variability in particle geometry in the SPC14 core, we utilize Dynamic Particle

Imaging (DPI) to measure particle shape properties (length, width, and aspect ratio [width/length]). DPI via Flow Cytometer and Microscope (FlowCAM) is a technique that allows for rapid counting, imaging, and measurement of particles (Sieracki et al., 1998) and is advantageous for ice core microparticle analyses because of its direct imaging capabilities and fast data output. DPI techniques have been extensively used in the biology community to analyze plankton species in a variety of environments and now by the ice core community (Álvarez et al., 2011; Maffezzoli et al., 2022). We use DPI at selected

depth intervals in the SPC14 core to obtain aspect ratio measurements, and then apply these data to continuous Abakus and

discrete CC measurements in order to assess interpretations of the SPC14 microparticle records. Depths were selected based on millennial scale Abakus dust concentration and size variability recorded in the preliminary SPC14 ice core. In addition, we combine DPI and Abakus data to calculate particle volumes and compare them against published particle shape calibration techniques from Simonsen et al. (2018). Our analyses provide the first millennially resolved time series of

particle shapes and SPC14 microparticle data spanning the interval 50 – 10 ka, providing valuable information regarding particle shape variability during the last glacial period and the start of the last termination in Antarctica.

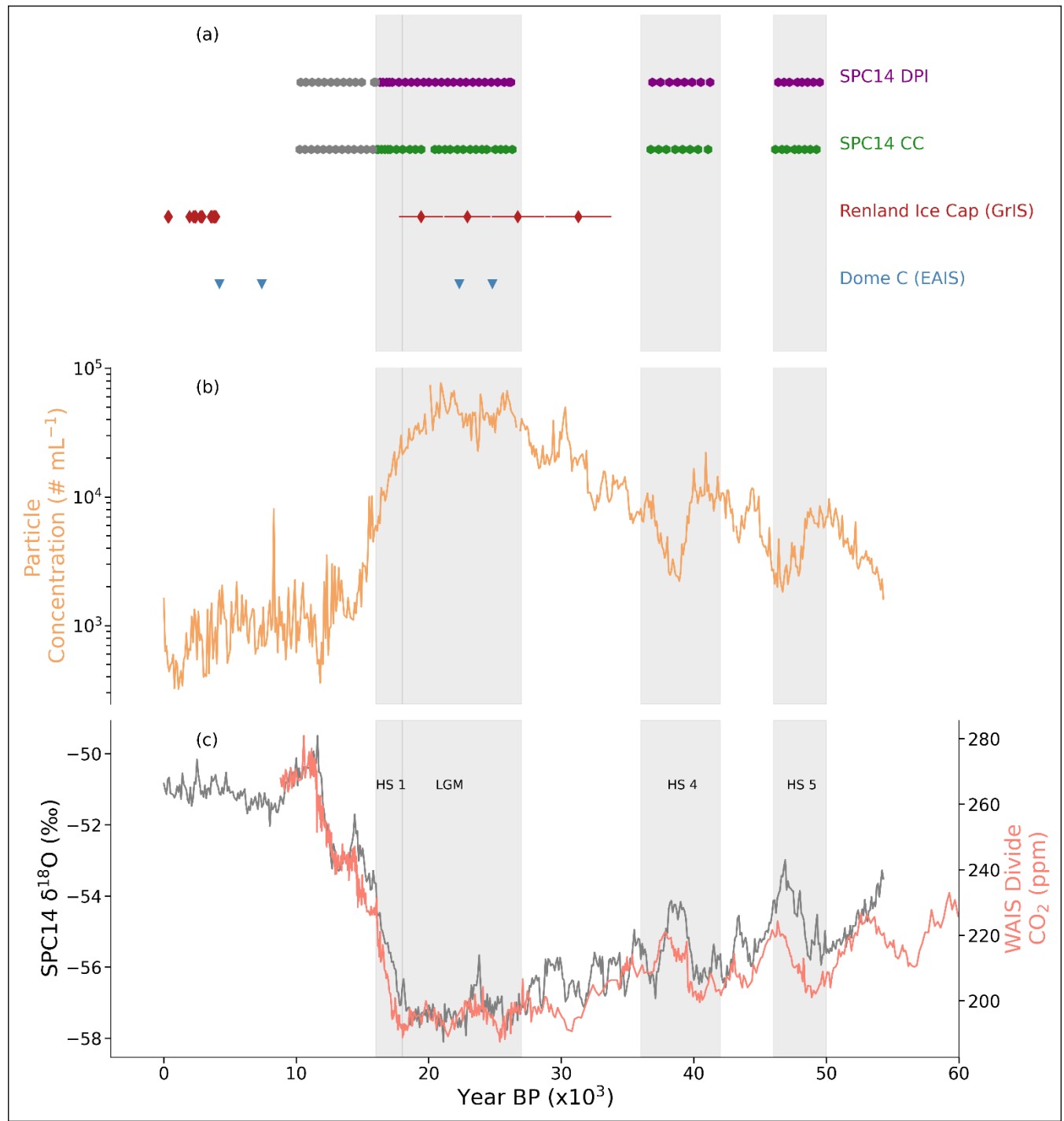

**Figure 1a - c: 1a) Ages of discrete particle shape analyses in this study compared to previously published research on particle shape variability during the Last Glacial Period (LGP): Potenza et al. (2016; Dome C, Antarctica; blue); Simonsen et al., (2018;**


Renland Ice Cap, Greenland; red); SPC14 CC (present study; green); and SPC14 DPI (present study; purple). Grey SPC14 CC and DPI markers represent samples that were run (see Methods). 1b) SPC14 dust particle concentration (present study; light brown), and 1c) $\delta^{18}O$ from Steig et al. (2021; grey), and WAIS Divide $CO_2$ record from Bauska et al. (2021; salmon) are shown for context. Grey bars highlight time periods of interest in our study.

## 2 Methods

### 2.1 Ice core recovery

The SPC14 core was drilled by the U.S. Ice Drilling Program with the Intermediate Depth Drill (IDD) using a 98 mm drill head (Winski et al., 2019; Johnson, 2014; Epifanio et al., 2020). Drilling fluid (Estisol-140) was introduced after reaching a depth of 160 m to reduce frictional resistance and to keep the borehole open. Core sections from the brittle ice zone (BIZ; 585 – 1077 m) were stored at the site for one year after drilling to depressurize before shipment to the U.S. Core depths, weight, and density measurements were completed at the U.S. National Science Foundation Ice Core Facility (NSF-ICF) in Denver, Colorado. Sections of the SPC14 core were cut into 2.4 x 2.4 x 100 cm 'sticks' at NSF-ICF and shipped frozen to Dartmouth College.

### 2.2 Abakus measurements

High-resolution Abakus measurements were made at Dartmouth College using a continuous flow analysis (CFA) melter system (Osterberg et al., 2006; Breton et al., 2012). Core sticks were melted on a 99.9995% pure chemical-vapor-deposited silicon carbide (CVD-SIC) melt head. The Dartmouth College CFA system has an effective resolution of 3 mm with signal dispersion lead of ~1 cm (Breton et al., 2012; Winski et al., 2019; Koffman et al., 2014; Osterberg et al., 2006). Microparticles were measured using a Klotz Abakus laser particle detector (Abakus). Peristaltic pumps drew melt water from the ice core through tubing directly into the Abakus sensor. Abakus particle concentrations (particles $\mu L^{-1}$) were calculated by dividing the Abakus output (particle count per unit time; usually 6 seconds) by the melt flow rates which were measured using a flowmeter (Sensirion SLI-2000). Twenty-four Abakus size bins ranged continuously in 0.1µm increments from endpoints (lower) 1.0 – 2.5 µm, followed by bins 2.7, 2.9, 3.2, 3.6, 4.0, 4.5, 5.1, 5.7, and 6.4 µm. Particle mass concentrations are estimated using an assumed density of 2.6 g $cm^{-3}$ (i.e., following Koffman et al., 2014). We calculate mass and flux measurements using spherical and ellipsoidal volume using measurements obtained from DPI measurements (see Section 2.4 and 2.5). Age data were interpolated based on the SP19 timescale (Epifanio et al., 2020; Winski et al., 2019). The Abakus was calibrated using CC techniques and then successfully tested against latex spheres of 1, 2, 5, and 10 µm diameter to assess accuracy (Koffman et al., 2014). Therefore, our size range of 1.1 – 6.4 µm have been successfully used and are considered accurate.

To check if the coarse particle concentration measured by the Abakus is composed of large particles rather than multiple small coagulated particles, we tested for coincidence following methods of Saey (1998), and used in Simonsen et al.

(2018). Saey (1998) showed that during measurement of ice with high particle concentration (>240,000 particles mL$^{-1}$) two particles passing through the detector at the same time can be erroneously identified as one large particle (hereafter referred to as 'particle coincidence'). Following methods of Simonsen et al. (2018), we compared the ratio of two different metrics of coarse particles (5.1 – 6.4 µm and 3.2 – 6.4 µm) to fine particles (<5.1 µm and <3.2 µm, respectively) against particle concentration during our periods of highest Abakus particle concentration, using a Spearman correlation test. If there is a high correlation of coarse particles to particle concentration, we would infer that particle coincidence occurred during CFA data collection. While we are aware of particle aggregation in deep sections of ice cores, we also access the impact of aggregation on PSD values. Baccolo et al. (2018) identified particle aggregates in the deep sections of the Talos Dome core PSDs which were characterized by low fine particle concentration and a coarse (~4.4 µm) mode. We look for similar features in the deep South Pole Ice Core.

### 2.2.1 Abakus data cleaning

Previous studies have highlighted the contaminating effect of the drilling fluid Estisol-140 on ice core microparticle concentrations (Warming et al., 2013). Estisol-140 peaks are characterized in the microparticle concentration data by exponentially increasing particle concentration followed by a sharp, several orders of magnitude decrease back to background concentration (Figure S1). Because of the repeatable nature of each Estisol-140 contamination peak, we were able to remove spurious data from our high-resolution Abakus microparticle dataset using a Python-based cleaning code. The code uses changes in coarse particle percentage (CPP) that overlap with changes in the rolling (3 point) median. If the metrics overlap, the data points in those time periods are flagged for removal. The cleaning code also utilizes the high-resolution Abakus data as well as multiple metrics collected in-line during CFA melting (i.e., electrical conductivity measurements and flow rates) and targets sections of the core where issues occurred during CFA melting and data collection and where erroneous peaks occurred from contamination of Estisol-140. In total, this approach indicates that, at most, 3% of the SPC14 record contains microparticle data affected by Estisol-140 (Figure S1). Similar quality control cannot be performed on the CC data because there are only 54 CC samples (see next section) compared with several hundred thousand Abakus measurements.

### 2.3 Coulter Counter measurements

During melting on the CFA system, discrete samples were collected every 2 meters (866 samples total; ~125 years per sample age span during the LGM) for glaciochemical analysis. Of these, we selected 106 to combine into 54 low-resolution (2 – 4 meters; see Supplementary Data) CC samples for comparison with the Abakus data (Figure 1; Supplementary Material). CC samples were analyzed at Colby College using a Beckman Coulter Multisizer 3 in a HEPA-filtered clean lab. To prepare a sample for analysis, we poured 10 ml from each of the 54 samples into cuvettes that were cleaned with a 5% Citranox solution and multiple rinses with Milli-Q™ ultrapure water. The samples were then mixed with Isoton II™ diluent (electrolyte solution) and inverted several times prior to analysis. Samples were measured using a 50 µm

aperture tube, yielding measurements of particle concentration across 300 size bins spanning 1.1 – 30 µm. Three-to-five

blank measurements were made at the start of each sampling day. Following clean blank runs, three-to-five measurements

were made for each of the 54 samples. Average blank measurements were about 160 times lower than raw CC data and were

significantly different from sample measurements according to a Student's t-test. We performed blank corrections by

subtracting the averaged median blank value from the median of each CC sample, since blank values are within $2\sigma$ of each

other. Because the CC produces more size bins than the Abakus (300 vs. 30 bins), we averaged CC particle size bin

concentrations to match Abakus bin sizes for direct comparisons.

## 2.4 Dynamic particle imaging

The FlowCAM (Yokogawa Fluid Imaging Technologies, Inc.) instrument at the University of Maine uses DPI to

measure particle length, width, and aspect ratios. DPI via FlowCAM was originally developed to photographically measure

phytoplankton or other particles 20-200 µm in length in concentrations ranging from $10^1 – 10^5$ L$^{-1}$ (Sieracki et al., 1998).

Modern DPI techniques using a 20x zoom can accurately image particles ≥2.0 µm in length, while recording particle shape

(length and width) information throughout each sampling run. Particle length, width, and aspect ratio measurements were

calculated using Feret measurements, defined as the perpendicular distance between parallel tangents on opposite sides of

the particles (Fluid Imaging Technologies, 2011). These measurements are taken 36 times in 5° angle increments from -90°

to +90°. The maximum distance recorded is the particle length and the minimum is recorded as the particle width. Aspect

ratios are then calculated by the width (*b* axis)/ length (*a* axis). Values range between 0–1, where 0 represents elongated

(oblate/prolate) particles and 1 represents a spherical particle. While 2D imaging does provide a length and width

measurement, this technique is limited because we cannot measure the third particle axis length (i.e., depth/height; *c* axis).

Aspect ratio measurements were grouped by length values into bins matching the Abakus measurements (i.e., 30 size bins;

section 2.2). We use the default factory settings, which have a predetermined calibration factor, which converts pixels to

micron measurements (Fluid Imaging Technologies, 2011).

We checked the quality of DPI data output by assessing particle image clarity and particle length measurements

using the edge gradient and particle-length measurement metrics. Because Feret measurements use the particle image to

identify particle length and width, images with poor clarity could introduce measurement errors. Therefore, we assess error

within DPI measurements via image clarity, instrument limits, and previously published error assessment of particle aspect

ratio measurements. More than 44,000 particles in total were measured throughout the sampling process with a minimum

particle length of 2 µm and minimum estimated diameter of 1 µm (see Figure S2 for particle counts by coarse particles (5.1 –

6.4 µm and total counts and Figure S3 for representative FlowCAM particle images from a single sample). The FlowCAM

automated output generates three estimates of particle diameter based on; 1) diameters based on the area of a circle, 2) mean

value of each Feret measurement, 3) weighted values of both method 1 and 2 (FlowCAM user manual).

We analyzed 54 samples via FlowCAM that were adjacent in depth to those measured on the CC (Figure 1a). Blank

measurements were collected at the start of each day and also between samples to ensure no background contamination; a

total of 141 blank measurements were collected. Blanks were comprised of deionized water (DI) that was run for 10 minutes at a flow rate of 0.04 – 0.07 mL min$^{-1}$. Prior to each run, the funnel and tubing were rinsed with a Citranox and DI solution, followed by a DI wash. Samples were then shaken and poured into the cleaned funnel and tubing. Samples were allowed to flow through a 0.5 mL cell prior to data collection to ensure that measurements were related to samples rather than the DI wash. The FlowCAM does not have the same size bin measurements as the Abakus and does not produce measurements for particle lengths between 2.1, 2.2, 2.4, and 2.7 µm. Because of the relatively similar mean widths and neighboring standard deviations, we used linear interpolation for these bin sizes to identify particle width (Figure S4 and Figure S5). Due to low particle counts (5.1 – 6.4 µm) in samples from 16 – 10 ka (Figure S2), we focus our analysis hereafter on the 41 samples older than 16 ka (Figure 1a). Although particle concentrations are high, we find no evidence of particle coincidence during HS1 or the LGM via our four checks of coincidence (-0.1 < r-values < 0.1, and with p-values <0.01 and > 0.01; Table S1). Following the methods and statistical tests for particle coincidence developed by (Saey, 1998) and used by Simonsen et al. (2018), the low r-value and p-value statistics indicate that while there is a significant relationship between the ratio of coarse to fine particles and particle concentration, this explains very little of the observed methodological offset. Therefore, because there is no relationship between the ratio of coarse to fine particles and particle concentration, these results suggest that the overestimation of coarse particles in the PSD$_{Abakus}$ is related to assumptions of particle shape/sphericity rather than data collection processes. If the mismatch between the Abakus and CC was related to coincidence, then we would expect a significant relationship between the two parameters. Furthermore, we find that the deeper core samples (Heinrich Stadial 4 and 5) PSD mode values compared to Heinrich Stadial 1 and LGM mode values are statistically similar (student t-test; p-value > 0.05). Therefore, while there maybe particle aggregation, it is not having a significant effect on the deep South Pole samples.

## 2.5 Abakus PSD calculation and calibration techniques

The main goal of this study is to explore the use of DPI and CC data to improve Abakus particle volume calculations. To this end, we used an array of methods to compare particle size metrics in Abakus data with concurrent CC and DPI samples. We compared four different methods for calculating particle volume and particle size distribution (PSD$_{Abakus}$) using a combination of Abakus, CC and DPI data, each of which are discussed briefly below. The resulting particle volume, mass concentration, flux, and PSD$_{Abakus}$ from each method are then compared to the corresponding CC particle volume, mass concentration, flux, and PSD$_{CC}$, which are assumed to be the most accurate method because the CC measures particle volume based on electrical impedance rather than through geometric calculations.

Aspect ratio data provide a direct test of the common assumption that ice core dust particles are spherical (i.e., Koffman et al., 2014; Ruth et al., 2003; Wegner et al., 2015) and allow us to calculate the influence of particle shape on a range of particle metrics. We apply our measured aspect ratios to the Abakus data using Eq. (1; below). Because the FlowCam captures a 2-dimensional snapshot of a 3-dimensional particle, two end-member calculations bracket the range of

true particle volumes for a given aspect ratio (b/a; Figure 2a and c). While we cannot measure the orientation of the particle in 3D space, the FlowCAM automatically measures the longest axis as the particle length (*a* axis) and the shortest axis as the particle width (*b* axis). Therefore, to account for particle height (*c* axis), we must assume that the height and/or depth of the particle is equal to either the a or b axis (Figure 2). The assumption that the c axis is equal to the a or b axis results in volumetric changes (Figure 2). By varying our height dimension, we can account for the missing particle dimension. We assume the following dimensions for each volumetric calculation; prolate assumes particle width (*b* axis) = height (*c* axis; Figure 2a), oblate assumes particle length (*a* axis) = height (*c* axis; Figure 2b), and sphere assumes all lengths equal to the particle length (*a* axis; Figure 2c).

$$Ellipsoidal\ 1\ Volume = \frac{4}{3}\pi(ab^2), \tag{1a}$$

$$Ellipsoidal\ 2\ Volume = \frac{4}{3}\pi(a^2b), \tag{1b}$$

$$Spherical\ Volume = \frac{4}{3}\pi r^3, \tag{1c}$$

In all cases, DPI measurements have been binned by particle length under the assumption that length measurements of particles are equal between the Abakus and DPI. While we recognize that the Abakus may be recording extinction cross section rather than true particle length, we assume that both measurement techniques (Abakus and DPI) are representing accurate particle length since the Abakus was calibrated to CC measurements (Koffman et al., 2014). Based on the dimension assumptions above, prolate represents a minimum volume, oblate represents a moderate volume, and sphere represents the null hypothesis of equal particle dimensions (maximum volume). Thus, three different calculation methods, corresponding to Eq. (1a, 1b, and 1c), are applied to generate three distinct datasets from the Abakus measurements, each with a corresponding volume, mass concentration, flux, and particle size distribution. For spherical volume r = ½ particle diameter. For ellipsoidal volume, a = ½ particle length and b = ½ particle width, c axis (particle height) is set to equal particle width (b-axis; prolate) or particle length (a-axis; oblate).

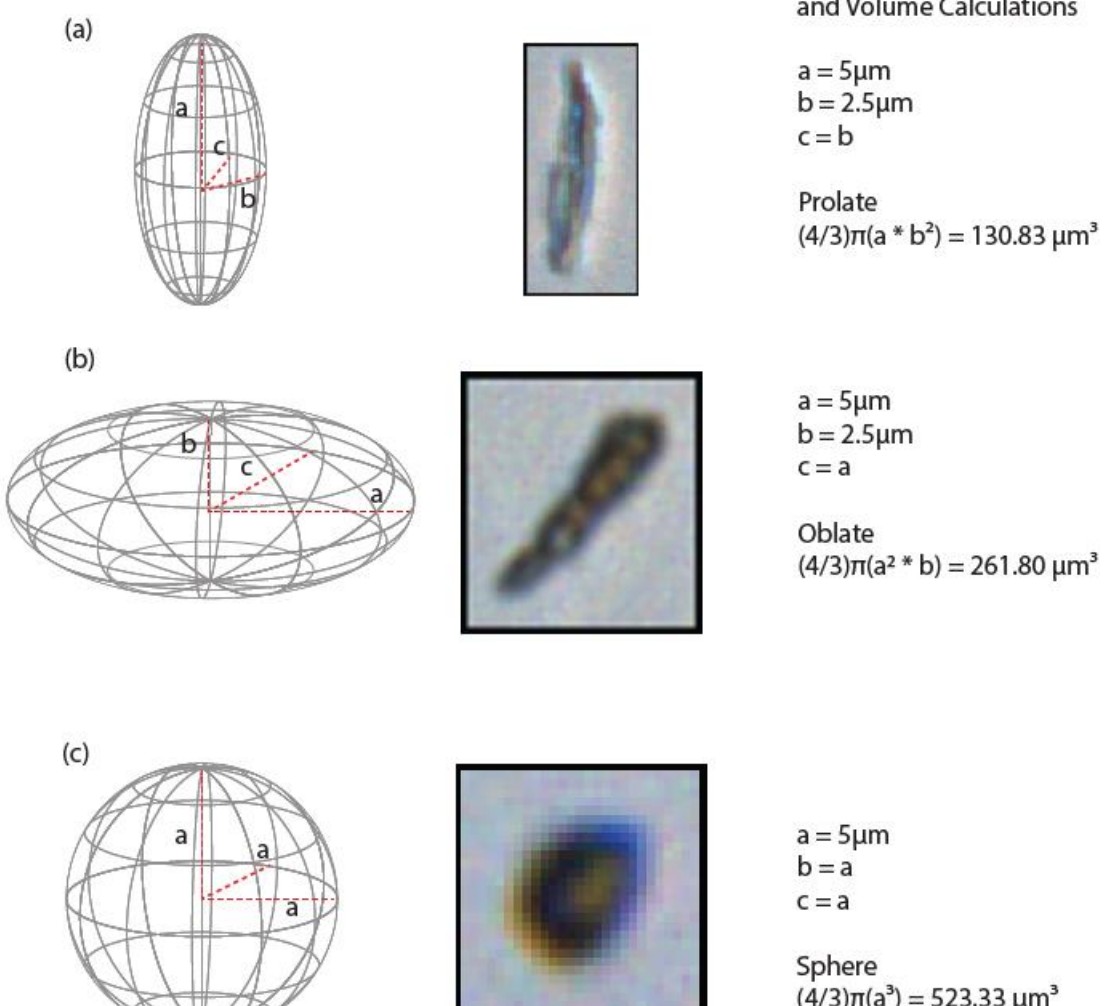

**Hypothesized Dimensions and Volume Calculations**

(a)

$a = 5\mu m$
$b = 2.5\mu m$
$c = b$

Prolate
$(4/3)\pi(a * b^2) = 130.83\ \mu m^3$

(b)

$a = 5\mu m$
$b = 2.5\mu m$
$c = a$

Oblate
$(4/3)\pi(a^2 * b) = 261.80\ \mu m^3$

(c)

$a = 5\mu m$
$b = a$
$c = a$

Sphere
$(4/3)\pi(a^3) = 523.33\ \mu m^3$

**Figure 2: Representative ellipsoidal and spherical particles with labelled axes showing: a) Prolate ellipsoidal particle shape with a FlowCAM image of a SPC14 particle; b) Oblate particle with rotated image of particle from a) (because particle images are in 2D we cannot differentiate between oblate and prolate); c) Spherical particle shape with a FlowCAM image of a SPC14 particle. Left-hand images are representations of hypothesized axis measurements and geometries. For each particle, the assumptions and geometric volume calculations are provided. The volume calculations highlight the large difference in particle volume between equal and unequal axes (Eq. 1).**

$$Adjusted\ Abakus\ size\ bins = \sqrt[3]{\bar{x}} * Abakus\ size\ bins, \tag{2}$$

$$A_N = c_i * Abakus\ size\ bins, \tag{3a}$$

$$Adjusted\ Abakus\ size\ bins = \min(|\ln(A_N) - \ln(CC_{bins})|), \qquad\qquad (3b)$$

In addition to the direct comparisons of particle metrics, we also explore methods that statistically adjust the Abakus data to match the more accurate CC data. We use two techniques developed by Simonsen et al. (2018) that were specifically designed to bring $PSD_{Abakus}$ in line with $PSD_{CC}$. The calibration techniques shift $PSD_{Abakus}$ using a spherical particle volume by corresponding averaged size aspect ratios and corresponding $PSD_{CC}$ (described below). In the first calibration method, we multiply the cubic root of the median aspect ratio for bin sizes 2.0 – 6.4 µm by the original Abakus

bin sizes (Eq. 2), generating a new set of $PSD_{Abakus}$ values. The second technique seeks to stretch, contract, and/or shift the calculated $PSD_{Abakus}$ to more closely match $PSD_{CC}$ following methods outlined by Simonsen et al. (2018). Here, we multiply values from the $PSD_{Abakus}$ by scalars ranging from 0.55 – 1.10 and then optimize the match with the $PSD_{CC}$ by minimizing the difference in the absolute value of the natural log of $PSD_{Abakus}$ and $PSD_{CC}$ (Eq. 3a and b). In Eq. (3a and b), $c_i$ is equal to all values ranging from 0.55 – 1.10 in increments of 0.01, $A_N$ = linearly scaled Abakus size bins, and $CC_{bins}$ = Coulter

Counter size bins. The applied shift results in the calibrated $PSD_{Abakus}$. While our study focuses on the change in shape based on 2D particle dimensions, we discuss the comparisons to both calibration techniques (Eq. 2, 3a, and 3b) in section 4.3.

    We used a combination of the above techniques (Eq. 1, 2, and 3a and 3b) to generate 7 sets of Abakus data and their associated metrics. For clarity, we categorize our analyses into calculations (Eq 1a, b, and c) and calibrations (Eq. 2 and 3a and 3b), where calculations refer to simple geometric differences (i.e., spherical vs ellipsoidal volume difference related to changes in shape), and calibrations refer to shifts or adjustments in Abakus data relative to CC data. Eq. 1 calculates particle

volume using three different particle axis dimensions. Eq. 2 shifts the $PSD_{Abakus}$ values based on aspect ratio measurements. We do not use Eq. 2 on ellipsoid $PSD_{Abakus}$ values because they already incorporate aspect ratios into their calculations. Finally, Eq. 3a and 3b manually shifts the $PSD_{Abakus}$ values to best fit $PSD_{CC}$. Figure 2 shows a depiction of the different hypothesized particle shapes and the resulting volume differences from changes in their geometries.


$$Total\ Offset = \sum_{i=size\ bin\ \geq 2.0}^{6.4} |(Abakus_i/CC_i) - 1|, \qquad\qquad (4)$$

    To summarize the difference between each methodology, we use Eq. (4), which we refer to as "total offset." Here, we assume that the particle sizes measured by the CC are the best representation of the true particle sizes in the ice core.

Conceptually, the total offset metric provides a summary statistic integrating the mismatch between Abakus and CC data across bins. This allows us to assess the various methodologies we apply to correct the Abakus output for non-spheroidal particle geometries. The total offset compares the ratio of the $d$Volume/$d$ln(Diameter) Abakus to the CC, the metrics used to assess particle size distributions in ice cores following Ruth et al. (2003). A value of 1 is subtracted from the ratio of Abakus: CC to shift an equal value to 0. The total offset, calculated for Abakus data, measures the sum of the differences from a 1:1

ratio between Abakus and CC PSDs using Abakus-defined bin sizes. The $Abakus_i/CC_i$ describes the Abakus median

calculation or calibration per size bin relative to the median CC sizes of similar size bins. In summary, based on this calculation, the lower the total offset the closer the two parameters are to each other.

## 3 Results

**3.1 Abakus and Coulter Counter measurement relationships**

Abakus and CC mass concentrations are positively and significantly correlated (Figure S6; r = 0.95, p-value < 0.001, linear slope$_{mass}$ = 2.32), consistent with the findings of Ruth et al. (2008, i.e., R$_{log}$ = 1.00). However, the Abakus mass concentration (using the assumption of spherical shape) is nearly twofold greater (~2.3) than that derived from the CC (Figure S6) and the average PSD$_{Abakus}$ mode values for all 41 samples is 4.80 ± 2.42 µm (2σ), while mean PSD$_{CC}$ is 2.70 ±

2.73 µm (2σ), suggesting that the Abakus over-estimates particle mass (i.e., volume). Abakus: CC PSD ratios (Figure S6 and Figure S7) show that while finer particles (<3.0µm) are closer to a 1:1 scale, coarser particles (3.0-5.0µm) are elevated (Abakus: CC >2), which indicates that coarse particles account for the bulk of the methodological offset (Figure S7). This non-linear relationship between fine and coarse particles measured in the Abakus and CC demonstrates that methodological variability needs to be accounted for. Furthermore, the distribution differences between PSD$_{Abakus}$ (under spherical shape

assumption) and PSD$_{CC}$ are not temporally consistent (Figure 3; Figure S7 and S8), potentially indicating changes in particle shape over time. Specifically, PSD$_{Abakus}$ (under spherical assumption) and PSD$_{CC}$ are most similar during HS 4 and 5 (total offset = 13.02; Table 1; Figure 3c and f), followed by the LGM (total offset = 22.16; Table 1; Figure 3b and e), and HS1 (total offset = 38.82; Table 1; Figure 3a and d). Across all time periods (41 samples representing 34 ka), the Abakus measures greater numbers of coarse particles relative to the CC, with the largest overestimation during HS1 (Figure 3a and

d). While particle sizes of 3 – 5 µm have the highest offset, the finest particles <2.0 µm throughout all time periods have similar apparent PSD ratios produced by Abakus and CC (Figure 3a – f).

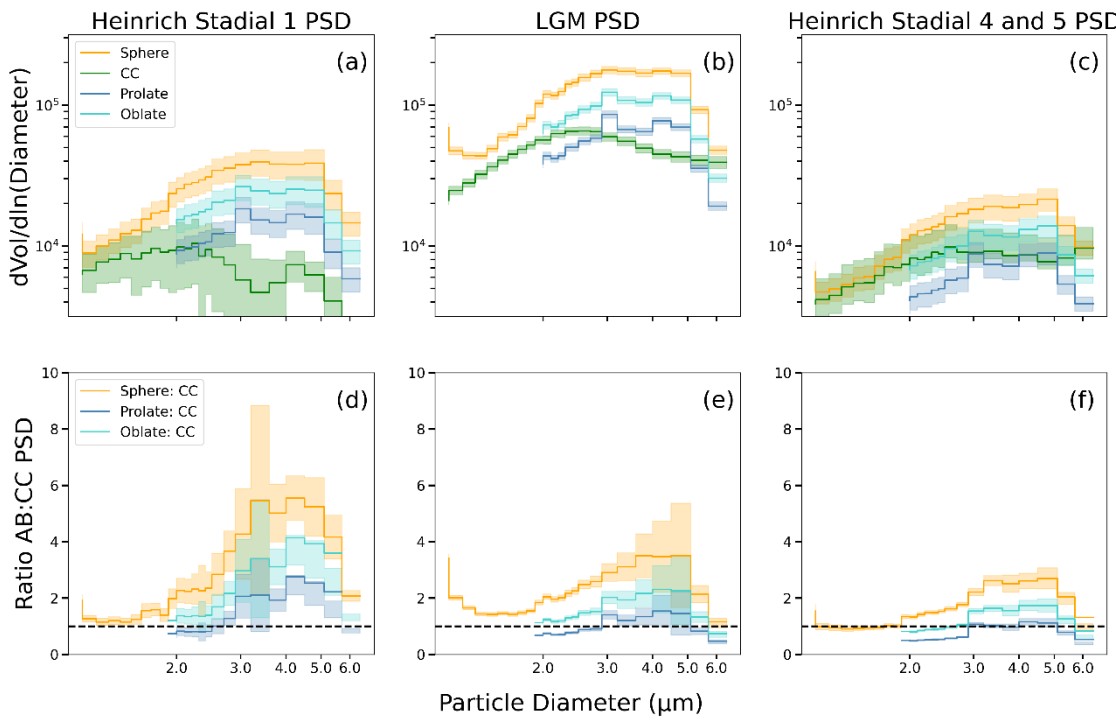

**Figure 3a – f: Comparison of Abakus volume and calibration schemes (Eq. 1, 2, and 3) to Coulter Counter data (top) and the ratio of particle size distributions between Abakus and Coulter Counter data (bottom). Sphere = Abakus sphere PSD Calculation (length = width = height), CC = CC PSD Calculation (accurate volume), prolate = Abakus prolate PSD calculation (width = height), Oblate = Abakus Oblate PSD calculation (length = height). The top panel compares the PSD$_{Abakus}$ volume and calibration techniques to PSD$_{CC}$ and the bottom panel is the ratio of each PSD$_{Abakus}$ to the PSD$_{CC}$ in different time periods. The dotted line in the lower panels represents a 1:1 value and the closer each colored line is to the 1:1 line, the lower the total offset. Using ellipsoidal volume calculations always improves methodological correspondence, although type of ellipsoid that best fits the data depends on particle size. For a comparison of all calculations and calibrations see Figure S9.**

**Table 1: Total offset measures for each calibration methodology relative to PSD$_{CC}$ values. Total offset is minimized for prolate CC Calibration, making it the preferred approach for this dataset.**

|  | HS1 | Glacial | HS 4 and 5 | Total Offset |
|---|---|---|---|---|
| Sphere PSD Calculation | 38.82 | 22.16 | 13.02 | 73.99 |

| | | | | |
|---|---|---|---|---|
| Prolate PSD Calculation | 9.11 | 4.49 | 4.81 | 18.41 |
| Oblate PSD Calculation | 19.02 | 8.64 | 4.56 | 32.22 |
| Aspect Ratio | 33.15 | 19.26 | 11.51 | 63.92 |
| Abakus Spherical CC Calibration | 9.21 | 22.03 | 12.12 | 43.36 |
| Abakus Prolate CC Calibration | 1.53 | 1.63 | 6.64 | 9.80 |
| Abakus Oblate CC Calibration | 2.31 | 8.89 | 19.23 | 30.43 |

## 3.2 Dynamic particle imaging

All size-dependent aspect ratios (width/length = b-axis/a-axis) have median values below 0.95, indicating the majority of particles are ellipsoidal (elongated) and not spherical (Figures 3a and c; Figure S10). Size-dependent aspect ratio
measurements are defined by two groups; 1) finer aspect ratios (< 5 µm) and 2) coarser aspect ratios (≥5.1 µm; Figure S10). Fine aspect ratios have a narrow distribution, while coarser aspect ratios have a broader distribution and are more elongated (Figure S10 and S11). Fine particles have median aspect ratios of $0.76 \pm 0.10$ ($2\sigma$). Coarse particles have a median aspect ratio of $0.70 \pm 0.14$ ($2\sigma$). Across all 41 samples, distributions of particle aspect ratio have a left skew towards elongated particles and are leptokurtic (Figure S10 and S11), indicating that along with a left skew, particles are more likely to be
asymmetrical rather than symmetrical (Figure 4a and c). There is no relationship between particle size-bins and distribution skewness (Figure 4a and c). While Mathaes et al. (2020) provided evidence of high error ranges in aspect ratio measurements using DPI at 10x for particles <5 µm, our aspect ratio measurements, obtained using a greater, 20x zoom (Figure 4b and d), had low and consistent aspect ratio standard deviations.

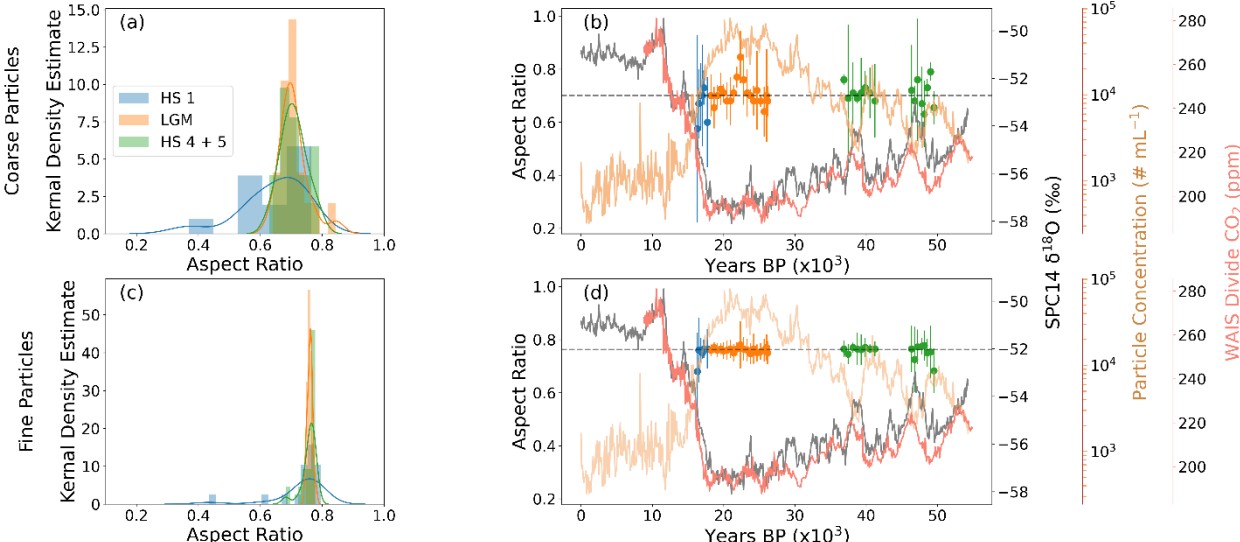

Figure 4 a – c: Particle aspect ratio data as a function of aspect ratio (a and c) and time (b and d). a and c) Aspect ratio distributions (medians) for HS1, LGM, and HS 4 and 5. 4b and d ) Median aspect ratio values (colors are related to time in panel a and c, respectively) with 2σ error bars plotted with SPC14 δ¹⁸O record (Steig et al., 2021; dark grey) and SPC14 particle concentration record (brown), WAIS Divide $CO_2$ (Bauska et al., 2021; salmon). The top two panels (a & b) show coarse particles (>5.0 μm) data and panels c & d show fine particles (<5.0 μm). The variability (2σ) increases in samples older than 18ka. Median aspect ratio values are identified by the black dashed line. Results show that aspect ratios are skewed towards more elongate values (< 1) with fine and coarse aspect ratios of 0.76 ± 0.5 and 0.70 ± 0.10 (2σ), respectively.

There are different temporal trends in the fine and coarse particle aspect ratios. Coarse particles are more variable through time and are slightly more elongated than fine particles (Figure 4b and d). Across the three time periods studied, we found coarse particle aspect ratios of 0.68 ± 0.12 (median ± 2σ, HS1), 0.70 ± 0.09 (LGM), and 0.71 ± 0.08 (HS 4 and 5). In comparison, fine particle aspect rates were 0.76 ± 0.07 (HS1), 0.76 ± 0.02 (LGM), and 0.77 ± 0.05 (HS 4 and 5). Using a t-test, we found that coarse and fine particle aspect ratios were significantly different for each period (p-value < 0.05). The variability of both fine and coarse particle aspect ratios appears to be higher during HS1 compared to the LGM and HS 4 and 5, though sample size may have contributed to these differences (HS1: n = 6, LGM: n = 19, and HS 4 and 5: n = 16).

Because of these differences in distribution and the non-linear offset between $PSD_{Abakus}$ and $PSD_{CC}$, we explore the differences between fine and coarse particles. For clarity, we refer to $PSD_{Abakus}$ and $PSD_{CC}$ when referring to the entire distribution of particles by size and fine and coarse particles when referencing generally trends based on distribution differences in aspect ratios.

Width measurements (equal to the bin size multiplied by the FlowCam aspect ratio) afford the opportunity to calculate non-spherical particle volume more accurately than under the assumption of equal width and height (prolate and oblate; Figure 2a and b). Width measurements ranged from 1.18 – 4.3 µm. We use linear interpolation for width measurements in bins 2.1, 2.2, 2.4, and 2.7 µm because of similarities in averages and 2σ for neighboring bin sizes (Figure S4 and S5). Averaged width measurements are not statistically different between each of the three time periods (Student's t-test, p-value > 0.1). Because of the temporally consistent average particle width measurements (HS1, 2.33 ± 1.87; LGM, 2.45 ± 2.07; HS4 and 5 2.40 ± 2.03; 2σ) for each bin in the SPC14 samples, we use average width measurements in particle volume calculations for the entire core and all dust particle metrics (Eq. 1; Figure S4 and S5). Therefore, we are assuming changes in aspect ratio are primarily driven by variations in particle length. We assess this assumption within the context of previously published measurements in Section 4.2.

### 3.3 Abakus calculation technique comparison (Eq. 1a, b, and c)

Following suggestions by Simonsen et al. (2018) we utilize the DPI results and evaluate two different metrics for calculating Abakus data. Calculated metrics include 1) spherical volume estimates (i.e., null hypothesis, Eq. 1c) and 2) ellipsoidal volume estimates (i.e., elongated shape hypothesis, Eq. 1a and 1b). During all time periods, the spherical $PSD_{Abakus}$ overestimates all particle sizes, having the greatest effect in the coarsest particles (Figure 3a - f and Table 1). During HS 4 and 5, oblate PSD calculation has the lowest total particle offset (closest to the CC) of all calculation and calibration techniques, which is followed by prolate PSD calculation (Table 1). Coarse particle $PSD_{Abakus}$ values calculated using either prolate or oblate (Eq. 1) volumes are closer to $PSD_{CC}$ values in all time periods compared to average spherical volumes (Figure 4b and d). Fine particle shape calculations do not have a consistent relationship to CC values. During HS1, prolate particle shapes closely match CC values, while during the LGM and HS 4 and 5, oblate particle shapes more closely match $PSD_{CC}$ values. Ellipsoid metrics produce the lowest total offset values for all time periods (Table 1). Of all three calculation metrics, prolate PSD calculations have the lowest total offset between HS1, LGM and HS 4 and 5, followed by oblate PSD. Ellipsoid calculations (both prolate and oblate) reduce total offset relative to spherical shape calculations by between ~25 – 44% (Table 1; Figure 3a – f).

### 4 Discussion

Calculating Abakus particle volume metrics using ellipsoid volumes reduces the discrepancy (offset) between Abakus and CC methodologies. Based on our results, we suggest there are three significant implications for Antarctic ice core microparticle analyses: 1) average particle shape is consistently ellipsoidal, not spherical, which likely represents microparticle mineralogy (e.g., clay particles and other sheet silicates); 2) assuming spherical particle shape leads to overestimation of coarse particle volume and/or mass; 3) using DPI, particle width measurements can be used to reduce observed offsets between Abakus and CC data; and 4) changes in particle shape through time (or a lack thereof) can be used

as an additional piece of information to assess changes in the dust cycle. The closer correspondence between ellipsoid values and PSD$_{CC}$ effectively reduces the magnitude of the Abakus offset that occurs when assuming a spherical particle shape (Figure 5). Younger than 16 ka, there is a discrepancy between the Abakus and CC values. We discuss the discrepancy in section 4.1 and the three implications of our microparticle analyses in section 4.2. While using ellipsoid volumes is important for reducing discrepancies among different microparticle measurement techniques, it also supports conclusions of

previous analyses of microparticle mineralogical composition and creates new avenues for future research, which are discussed in section 4.5.

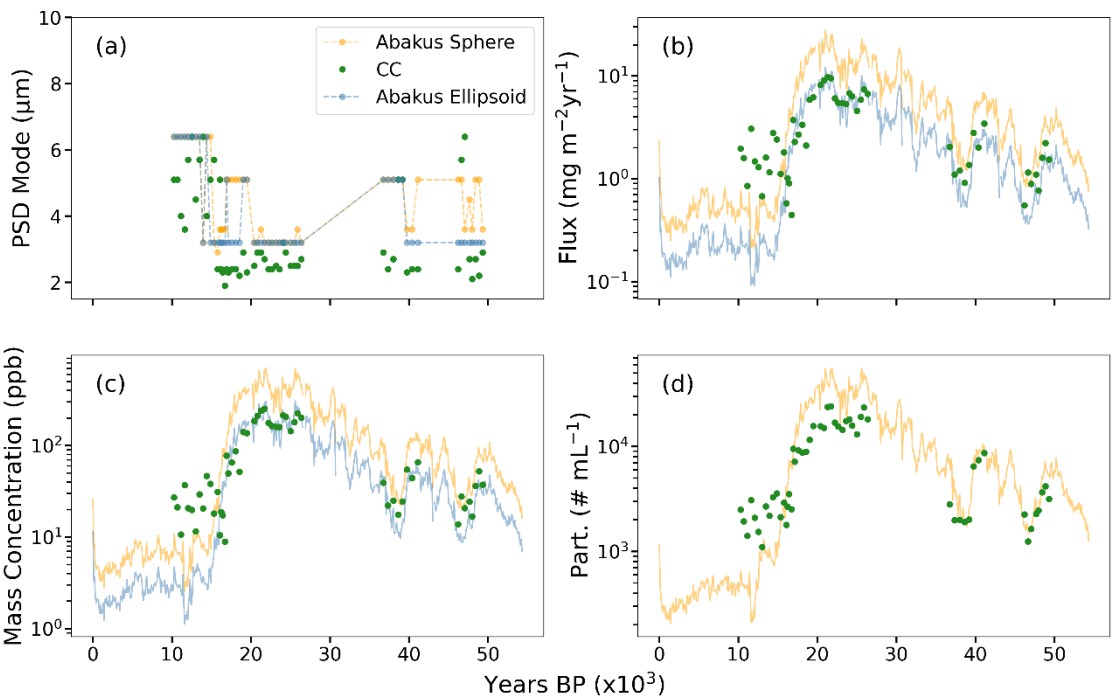

**Figure 5. Comparison of CC and Abakus spherical and ellipsoidal (prolate, Eq. 1b) particle data including a) volume mode, b) flux, c) mass concentration, and d) number concentration. The Abakus calculated prolate data are a better fit to the CC data for**

**all volume related metrics (flux, mass concentration (ppb), and PSD). Number concentrations (d) are unaffected by particle shape.**

### 4.1 Effect of Estisol-140

       During the last glacial period until 16 ka, there is a close correspondence between all metrics (mean and median resampled data) from the Abakus and the CC (Figure 5, Figure S12). However, Estisol-140 affects both the Abakus and CC

data from 16 – 0 ka because of the relatively low dust signal. The clearest example of this is a stepwise increase in particle

concentration with the introduction of the drilling fluid at 160 m (~2 ka; not shown). Because dust signals during the interval from 16 – 10 ka are 3 – 5 orders of magnitude lower than during the LGM, the influence of Estisol-140 rivals the true microparticle signal in the younger samples. While we are able to remove contamination peaks caused by drilling fluid in the continuous Abakus data, we are not able to separate these signals in the lower-resolution CC samples during times of low dust concentrations. Therefore, we do not interpret CC samples younger than 16 ka.

## 4.2 Ellipsoid particle shape

Our measurements (Figure 3, 4, 5 and Table 1) show that average particle shape is ellipsoidal, consistent with previous studies (Simonsen et al., 2018; Potenza et al., 2016). In SPC14, coarse and fine particle shapes are distributed around median aspect ratio values of 0.70 and 0.76, respectively. Using an ellipsoidal particle shape informed by PSD$_{Abakus}$: PSD$_{CC}$ measurements reconciles most of the total offset between Abakus and CC data (Table 1; Figure 5). Together with the FlowCAM images, these results confirm that microparticles are predominantly elongated rather than spherical. Prolate (Figure 2a) geometries have shorter width/height measurements compared to oblate or spherical geometries. Our assumption of consistent particle width measurements seems to be supported by our results (i.e., reduced discrepancy between Abakus and CC; Figure 3, 4, 5 and Table 1). Future studies should use this assumption with caution after measuring site specific particle metrics on the timescale in question. Mathaes et al. (2020) identified that measurement techniques can lead to greater variability of 5 µm spherical (aspect ratio = 1.0) or elongated particles (aspect ratio = 0.2). However, our aspect ratio measurements consistently cluster between values of 0.70 to 0.76 across a wide range of time periods and particle sizes (Figure S10). Not only are these aspect ratios outside the range of spherical and elongated particles (as defined by Mathaes et al. (2020)), but their consistency demonstrates the repeatability of our techniques. Despite our millennial-scale consistency in aspect ratios at the South Pole, Potenza et al. (2016) identified that during short decadal time periods (i.e., ~15 years), particle aspect ratio is variable, indicating that high-frequency changes in particle shape may exist despite the low-frequency stability. Furthermore, it is important to note that our samples occur during 50 – 16 ka, prior to interpreted atmospheric reorganization and changes in dust particle source area variability (Wegner et al., 2012; Aarons et al., 2017). Therefore, while our results under the assumption of assumed constant particle width measurements does reduce discrepancies between Abakus and CC volume related metrics, we advise caution to using this assumption without measuring particle shape at different temporal scales and spatial locations.

While most particles are consistently ellipsoidal, ellipsoid types (Figure 2a and b) are not consistent through time. During HS 4 and 5, oblate (Eq. 1c) provides a better fit to the CC data than prolate PSD (Eq. 1b), which has a better fit during HS1 and the LGM. While aspect ratio measurements remain relatively constant through time, it is likely that the third, hidden dimension of particle height (axis c) varies through time. PSD$_{CC}$ values lie between prolate and oblate PSD$_{Abakus}$ values indicating that our two ellipsoidal geometries represent bounds on volume for a given aspect ratio. This information could imply a shift in composition from prolate/linear to oblate/planar shaped particles between 36 – 27 ka, perhaps indicative of a change in source environment or mineralogy.

While a study of Saharan dust transport with a larger range of sizes (i.e., Von Holdt et al., 2021) identified particle
roundness ranging from 0.2-1.0 (aspect ratio) with varying complex mineralogy, our study shows relatively consistent and
invariant fine particles <5.0 µm with slightly more variable coarse particles. The consistency of South Pole particle aspect
ratios may be indicative of the distal continental source during the last glacial period, which we infer to be South America
based on geochemical analyses of dust from Vostok and Dome C (Delmonte et al., 2008; Delmonte et al., 2004). The slight
temporal variability in the coarser particles may indicate some mineralogical and/or source variability (e.g., Arnold et al.,
460 1998).

Particle mineralogy and shape in ice cores are mechanistically linked. Arnold et al. (1998) identified that particles
that are transported farther are smaller and tend to be composed of more platy/secondary minerals (i.e., clays). Particles that
were transported from closer dust sources tended to be coarser in size and composed of more primary minerals (i.e.,
plagioclase). This relationship was utilized by Paleari et al. (2019) to identify variations in source region between the
Holocene and the LGM at Dome C, Antarctica. Paleari et al. (2019) noted that although LGM and Holocene particles were
mineralogically similar, the Holocene contained more volcanic and metamorphic minerals, and those indicative of strong
weathering. Although they identified some mineralogical variability, they interpreted these results as an indication that there
had not been a significant shift in dust source region to Dome C between the LGM and Holocene. Prior research on the
mineralogy of modern dust at the South Pole indicates that particles are predominantly composed of clay minerals,
specifically, illite (20%), kaolinite (8%), halloysite (4%), vermiculite (3%), and other related clay minerals (24%; Kumai,
1976). The relatively small grain sizes coupled with predominant clay mineralogy at the South Pole is indicative of long-
range transport. Recent studies have also identified elongate diatoms in snow and ice samples from Dome C and Western
Antarctica, providing further evidence of the wide range of particle shapes in Antarctic aeolian samples (i.e., Delmonte et al.,
2017; Allen et al., 2020). While small clay particles (~2.5 µm) are predominantly oblate particle shapes, larger particles (~10
µm) became more variable in shape (Meland et al., 2012). The close correspondence between our $PSD_{Abakus}$ and $PSD_{CC}$ fine
particles supports their interpretation. The disagreement between our $PSD_{Abakus}$ and $PSD_{CC}$ coarse particle further suggests
that coarse particles are more variable in shape.

Particle shape can affect particle deposition and radiative properties (i.e., Li and Osada, 2007a, b; Von Holdt et al.,
2021; and references within; Knippertz and Stuut, 2014). For instance, rounder particles tend to settle out first, while
elongated particles remain in the atmosphere longer due to resistive forces (Ginoux, 2003; Formenti et al., 2011; and
references within). The gravitational center location on individual dust particles, particularly if asymmetric, can also
influence settling and transport Li and Osada (2007b). While particle shape can have negligible effects on light scattering, it
can impact aerosol optical depth calculations (a technique used to calculate atmospheric dust concentrations) by 30 – 40%
(Potenza et al., 2016; Feng et al., 2015).

While our data suggest that particle width is constant throughout our record, we acknowledge that this may be due
to our relatively low resolution of our analysis. For example, Potenza et al. (2016) measured aspect ratios of particles
recovered from EPICA Dome C, which varied between prolate and oblate within 10 – 15 years. Simonsen et al. (2018) also

state in their interpretations, that more extreme aspect ratios are more likely related to long-distance transport, where short-dust storms increase variability. Because we are measuring selected millennial-scale aspect ratio variability, this suggests

that our interpretations could reflect constant average dust source weathering and transport conditions to the South Pole, since particle shape can influence settling and transport (Li and Osada, 2007b; and references within; Ginoux, 2003; Formenti et al., 2011). We suspect that analyses of short-term variability would likely highlight more variability in particle width.

### 4.3 Particle size and shape effect on calculated volume

Our results indicate that the accuracy of certain Abakus data metrics can be improved by incorporating measurements of particle shape (i.e., aspect ratio). Incorporating aspect ratios reduced the total offsets by 25 – 44% (Table 1). While any volume-related Abakus particle metric requires calculation and possibly calibration using the DPI and CC techniques, the non-volume related metrics (i.e., particle number concentration) do not require calibrating to external methods. Particle number concentration showed no evidence, even during periods of high concentration, of coincident

counts. Our tests for particle coincidence indicate that there is either a weak but significant relationship (p-value < 0.01), or no relationship, between the ratio of coarse to fine particles and the particle concentration. Furthermore, the Abakus concentrations scaled to CC sample resolution (i.e., 54 discrete samples) and CC samples have a significantly strong correlation (r = 0.95, p-value <0.01; Figure S6), similar to relationships published in Ruth et al. (2008). Our results indicate that number concentration measurements derived from high-resolution particle Abakus measurements are accurate, in

agreement with previous microparticle methodological analyses (Ruth et al., 2008).

By using the ellipsoid volume calculations alone, differences between the Abakus and CC flux and mass concentration were reduced within each time period (Figure 3 and Figure 5b and c). Ellipsoidal volumes reduced Abakus median particle flux and mass concentration relative to the corresponding CC values from a ratio of 1.82 (with an assumed spherical volume) to 0.79 (with an assumed prolate volume) and 1.20 (with an assumed oblate volume, where a value of 1 is

equal). Without calculation/calibration using DPI and CC techniques, assumptions made while using Abakus volume metrics could overestimate mass and flux up to ~980% (i.e., HS1; Figure 3d). In agreement with previous studies (Simonsen et al., 2018; Potenza et al., 2016), we find that SPC14 particles are elongated rather than spherical, fitting somewhere between oblate and prolate particles (Figure 2 and 3; Potenza et al., 2016; Simonsen et al., 2018). Using prolate volumes compared to spherical volumes reduced average percent difference between the Abakus and CC by 221%, 147%, and 105% during HS1,

the LGM, and HS 4 and 5, respectively. Using oblate volumes reduced the average percent difference between Abakus and CC metrics by 130%, 89%, and 66% during HS1, the LGM, and HS 4 and 5, respectively. Percent difference values that are over 100% represent over-correction of the calculation/calibration method.

Particle mode is a useful measurement for analysis of varying PSD and has been used to identify changes in atmospheric structure (i.e., Koffman et al., 2014; Delmonte et al., 2017; Ruth et al., 2003). Although our calculation

techniques did lower mode particle diameter values overall, they are still within the 2σ envelopes of the original Abakus

spherical PSD calculation (4.80 ± 2.42, 3.20 ± 2.41, 3.20 ± 2.60; spherical PSD, prolate PSD, and oblate PSD mode values; Figure 5d). And although the median (average) PSD$_{Abakus}$ mode is reduced overall and closer to the CC PSD mode value (2.70 ± 2.74; 2σ), a t-test between each of the four metrics shows that spherical and oblate PSD are significantly similar (p-value > 0.05) while prolate and CC volumes are statistically different (p-value <0.05). While the high standard deviation in

mode values is high, these results suggest that prolate mode PSD values are the most similar to CC values.

### 4.4 DPI as a technique for particle calibration

DPI improved Abakus particle volume metric calculations and calibrations by providing width and aspect ratio measurements that refine calculations of particle volume. Our results highlight that not only are spherical particle shape assumptions incorrect but that their calibration techniques at the South Pole were less effective compared to calculation

and/or calibration techniques that used ellipsoidal shapes and volumes. Total offset values as well as coarse particle offset clearly indicate that the use of width measurements from DPI improved the fit of Abakus to CC data. Given sufficient DPI measurements to characterize any temporal or size dependent variability in particle shape, this technique could be used to calibrate Abakus volume, mass, and flux estimates in future ice coring studies. While DPI does not provide 3D measurements, the assumption of varying height to either particle length (oblate) or width (prolate) still resulted in decreased

discrepancy between the Abakus and CC (i.e., reduced total offset; Table 1). Measurements of particle shape also greatly improve understanding of particle size distribution in Abakus data. This non-linear offset between Abakus and CC could lead to potentially spurious interpretations of mode particle size, a commonly used metric for interpreting transport distance and wind strength (Koffman et al., 2014; Ruth et al., 2003; Delmonte et al., 2017; Steffensen, 1997). Use of DPI with previously published calibration techniques further reduces Abakus volume related metrics compared to similar CC data

(Simonsen et al., 2018; Potenza et al., 2016). We also note that the reduced discrepancy between the Abakus and CC data supports our hypothesis that DPI particle geometries can be matched with Abakus particle length (extinction cross section) measurements. If the Abakus geometric dimensions were extremely biased on millennial timescales, then we would expect the use of DPI to not reduce discrepancy between the two datasets.

We recommend the following steps for use of DPI as technique for more accurate volume-corrected Abakus particle

metric calibration: 1) select samples for DPI based on changes in Abakus concentration, flux and/or PSD, 2) adjust particle measurements to match Abakus bin sizes, 3) vary measurements to account for 3D variability (i.e. height value) between prolate, oblate, and spherical particle shapes, 4) compare new calculated particle size distributions using the total offset parameter as a metric for comparison between Abakus and CC data, 5) use the particle shape calculation that produces the lowest total offset between the Abakus and the CC.

### 4.5 Implications of South Pole particle shape in paleoclimate reconstructions

Temporal variations in particle shape may provide additional information on past climate, atmospheric, and environmental dynamics, because transport and depositional velocities, dust mineralogy, and transport distance all likely

have an influence on particle shape at a particular depositional location through time (Von Holdt et al., 2021; Knippertz and Stuut, 2014; Li and Osada, 2007b). Each of the time periods highlighted in this study (HS1, LGM, HS4 and HS5) represent

times of significant climate change based on the SPC14 $\delta^{18}O$ record (e.g., temperature) and atmospheric $CO_2$ variability (Figures 1 and 3). Dust transported into interior East Antarctica is predominantly from southern South America via the Southern Hemisphere Westerly Winds (Delmonte et al., 2008; Delmonte et al., 2020). Because of the consistency of particle shape and similarities between the Abakus volume corrected particle flux and mass metrics to similar measurements made on the CC, the calibrated SPC14 particle record can be used as a robust volume accurate metric for atmospheric dynamics

over the past 54,000 years. The consistency in particle shape further implies consistent mechanisms of transport.

**5 Conclusion**

We used dynamic particle imaging (DPI) to assess microparticle shape in the SPC14 record between 50 – 16 ka. Our results represent the longest and highest-resolution record of particle shape yet developed from ice cores. Through DPI we were able to: 1) use average particle dimensions to lower inaccuracies associated with shape assumptions, 2) generate

new techniques for calculating particle metrics from Abakus data, and 3) develop a new calibrated SPC14 Abakus dust dataset. Our comparison of SPC14 microparticle data from three different instruments and methodologies reveals consistently ellipsoid particle shapes, though with some temporal variations in inferred axis rotation (e.g., prolate vs. oblate). In agreement with findings from the Renland Ice Cap (Greenland; Simonsen et al., 2018), our results demonstrate that microparticles deposited in polar snow and ice are consistently oblong and not spherical. Assumptions of sphericity lead to

larger offsets between Abakus and CC data, greatly increasing volume-based metrics in the Abakus data. Our recommendations include the following steps for calculating best fit for Abakus-derived volumetric data: 1) Use DPI to obtain length and width measurements for particles, 2) calculate ellipsoidal volumes and PSD$_{Abakus}$ and use Simonsen et al. (2018) method 3 for adjustments, depending on the application, and 3) minimize the total offset parameter. We recommend for future dust volume calculations that a robust reconnaissance of dust particle shape be completed based on drill site and

dust profile characteristics (i.e., concentration, size, chemistry, and/or source region variability). The methods presented here produce a high-resolution continuous Abakus record that is most comparable to discrete CC measurements. The consistency in particle shape suggests that consistent transport mechanisms occurred between 50 – 16 ka.

**Code availability**

The Python-based code used to clean the SPC14 Abakus dataset is available at https://github.com/katherine-anderson/SPICEcore_Dust_Data_Processing (Andreson, 2020).

**Data availability**

Dust datasets used in this paper are available at the U.S. Antarctic Program Data Center (DOI: 10.15784/601553.).

## Author contribution

AC collected and analyzed the data and prepared the manuscript with contributions from all co-authors. KK, EO, JCD, and MW designed the experiments and guided the overall research effort. DF and ZT melted the core and measured the Abakus data and collected discrete samples. BK supported and provided laboratory for Coulter Counter measurements. JM supported FlowCAM measurements. KA wrote the Abakus cleaning code. DW provided data analysis and writing support. NH measured a subset of the Coulter Counter samples. AP provided writing support. MH provided laboratory analysis feedback.

## Competing interests

The authors declare that they have no conflict of interest.

## Acknowledgements

This research was funded by the US National Science Foundation grants [1443336 (Osterberg); 1443397 (Kreutz); 1443663 (Cole-Dai); and 1443105, 1141839 (Steig)]. We thank Mark Twickler, Joseph Souney and the South Pole Ice Core (SPICEcore) Science Coordination Office for administering the project; the U.S. Ice Drilling Program for support activities through NSF Cooperative Agreement 1836328; the 109[th] New York Air National Guard for airlift in Antarctica; the field team who helped collect the core; the members of South Pole station who facilitated the field operations; the National Science Foundation Ice Core Facility for ice core processing; and the many student researchers involved in producing the data used in this research. We thank Dr. Jasmine Saros and the Sawyer Water Research Laboratory for the use of the FlowCAM. We thank two anonymous reviewers, whose thoughtful suggestions and questions served to clarify and improve the paper.

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
