# Peer review of "Non-spherical microparticle shape in Antarctica during the last glacial period affects dust volume-related metrics"

_Climate of the Past, 2022_

## Author Comment (AC1)

We thank Reviewer 1 for their thoughtful and insightful comments.

*Major Comments 1: About the procedure through which the results are obtained. As suggested by the authors in the conclusion, shape should be considered for dust analysis and implications. Therefore, a reader could better exploit the results of this work if the calibration procedure will be more explicitly presented in the text.*

We add the following steps for a reader to follow for more accurate calculation of particle shape in Lines 545 – 550: "We recommend the following steps for use of DPI as a technique for more accurate volume-corrected Abakus particle metric calibration: 1) select samples for DPI based on changes in Abakus concentration, flux, and/or PSD, 2) adjust particle measurements to match Abakus bin sizes, 3) vary measurements to account for 3D variability (i.e. height value) between prolate, oblate, and spherical particle shapes, 4) compare new calculated particle size distributions using the total offset parameter as a metric for comparison between Abakus and CC data, 5) use the particle shape calculation that produces the lowest total offset between the Abakus and the CC."

*I find the explanation about "calculations" and "calibrations" and the description in sect. 3.3 maybe too short for a reader who wants to exploit the method.*

We add to our current definition the following description to our definitions of "calculations" and "calibrations" in line 289 – 291: "For clarity, we categorize our analyses into calculations (Eq 1a, b, and c) and calibrations (Eq. 2 and 3a and 3b), where calculations refer to simple geometric differences (i.e., spherical vs ellipsoidal volume difference related to changes in shape), and calibrations refer to shifts or adjustments in Abakus data relative to CC data."

*Major Comment 2: I find the discussion about the 2D images and the impossibility to differentiate between prolate and oblate to be correct but too short to be appreciated by a non-expert in the field.*

*More precisely:*

*Line 187: please introduce here the limitation imposed by the 2D measurement*

Line 187 (Line 192): We add the following sentence to introduce the limitation of 2D measurements, "While 2D imaging does provide a length and width measurement, this technique is limited because we cannot measure the third particle axis length (i.e., depth/height; $c$ axis)."

*Line 225: please discuss briefly this limitation in the text. As it is now I find the key point to be much more clear in the figure caption.*

Line 225 (now Line 241 - 245): We clarify our assumption of how we account for changes in particle height ($c$ axis) and variability in average particle shape by adding the following sentences, "While we cannot measure the orientation of the particle in 3D space, the FlowCAM automatically measures the longest axis as the particle length ($a$ axis) and the shortest axis as the particle width ($b$ axis). Therefore, to account for particle height ($c$ axis), we must assume that

the height and/or depth of the particle is equal to either the a or b axis (Figure 2). The assumption that the c axis is equal to the a or b axis results in volumetric changes (Figure 2). By varying our height dimension, we can account for the missing particle dimension."

*I think better to explicitly name "oblate" and "prolate" the shapes in the text.*

All Ellipsoid 1 and Ellipsoid 2 have been changed to "prolate" and "oblate".

*Figure 2: I suggest not to use the same image as an example of particles (a) and (b).  It generates confusion. I understand the author's idea, that I find very fair, but I  think it could be better explained insteada of showing the same image.*

Figure 2: We added new image of particle to Figure 2b for clarity and changed Ellipsoid 1 and 2 to prolate and oblate. Aspect ratio measurements calculated in Figure 2a, b, and c are hypothesized and do not reflect actual measurements of the particles, which is noted in the Figure 2 caption and now in the figure.

*A brief discussion of this issue could also be inserted in Sect 4.4 (line 467).*

Line 467 (now Line 534 - 536): We add the following discussion regarding our assumption of particle height, "While DPI does not provide 3D measurements, the assumption of varying height to either particle length (oblate) or width (prolate) still resulted in decreased discrepancy between the Abakus and CC (i.e., reduced total offset; Table 1)."

*On top of that, I respectfully suggest the authors to check for numbers a and b in Fig.2. The aspect ratio of the imaged particle is apparently much larger than 2. Could it be due to a bias in the method to obtain the Feret size from the images?*

The images showed in Figure 2 and their corresponding dimensions are hypothesized and used as an example rather than absolute values.  While the values in this figure are hypothesized which is stated in the caption, we also add "Hypothesized Dimensions and Volume Calculations" to the top of the figure for clarity.

*Major Comment 3: 3) I think something is wrong, or wrongly explained, about the measurement of "width" and the interpretation (lines 335-343).*

*Apparently, the authors are able to characterize particles better than the ellipsoidal description, but this is not really clear. Also the difference between width and height is not clear, because their definition should depend on orientation.*

*Moreover, the conclusion that "width measurements are not statistically different between each of the three time periods" at variance with the shapes, appears very peculiar. Also peculiar is that "changes in the aspect ratio are primarily driven by variations in particle length".*

*What I respectfully suspect is a kind of bias or oversimplification in this part of the analysis.*

*If "widths" would really be independent of time a specific careful interpretation should be given, or at least attempted on the basis of independent information. It would be an outstanding discovery indeed, motivating to re-consider all the distributions with the width as the key parameter to characterize dust.*

We add the following (Lines 241 – 248) for clarification of our particle height assumption, "While we cannot measure the orientation of the particle in 3D space, the FlowCAM automatically measures the longest axis as the particle length (*a* axis) and the shortest axis as the particle width (*b* axis). Therefore, to account for particle height (*c* axis), we must assume that the height and/or depth of the particle is equal to either the a or b axis (Figure 2). The assumption that the c axis is equal to the a or b axis results in volumetric changes (Figure 2). By varying our height dimension, we can account for the missing particle dimension. We assume the following dimensions for each volumetric calculation; prolate assumes particle width (*b* axis) = height (*c* axis; Figure 2a), oblate assumes particle length (*a* axis) = height (*c* axis; Figure 2b), and sphere assumes all lengths equal to the particle length (*a* axis; Figure 2c)."

We add the following discussion to Section 4.2 (Lines 486 - 494) to clarify our interpretation of our width measurements, "While our data suggest that particle width is constant throughout our record, we acknowledge that this may be due to our relatively low resolution of our analysis. For example, Potenza et al. (2016) measured aspect ratios of particles recovered from EPICA Dome C, which varied between prolate and oblate within 10 – 15 years. Simonsen et al. (2018) also state in their interpretations, that more extreme aspect ratios are more likely related to long-distance transport, where short-dust storms increase variability. Because we are measuring selected millennial-scale aspect ratio variability, this suggests that our interpretations could reflect constant average dust source weathering and transport conditions to the South Pole, since particle shape can influence settling and transport (Li and Osada, 2007b; and references within; Ginoux, 2003; Formenti et al., 2011). We suspect that analyses of short-term variability would likely highlight more variability in particle width."

We add that on millennial scales, particle median particle width measurements per size bin are statistically similar, therefore, we use averaged measurements for the entire record. While previous works (i.e., Potenza et al., 2016) identified that particles widths are variable during short time periods (< 15 years), this may be different over millennial scales, at least at the South Pole where aspect ratios change little between 50 – 16 ka. We also recognize that all of our dust particle measurements occur during the glacial period (54 – 16 ka), occurring before any major atmospheric reorganization that occurred during the Termination I (18 – 12 ka). Geochemical provenance data suggests that Antarctica dust particle source areas began increasing in variability between 16 – 15 ka (i.e., Wegner et al., 2012 and Aarons et al., 2018). We suggest that the statistically similar similarities in median particle width might reflect similarities in dust transport/source regions. Furthermore, this is the highest resolution record of particle length and width measurements. Therefore, this interpretation may be specific to the South Pole. As a first order/exploratory analysis, our results of reduced discrepancy between the Abakus and CC via the assumption of common dust particle width seems to be supported by our results.

*Minor Comments*

*line 68: Abakus does not measure backscattered light*

Line 68 (Line 67 – 70): Changed "backscattered light" to "The Abakus measures particles via disruption of transmitted light; as particles flow through the sensor, the interruption of the light source produces scattering and shadowing resulting in a negative peak in transmitted light, which is used to measure the diameter of the particle (Ruth et al., 2002)."

*line 72: "it only provides particle size measurements in one dimension". Please clarify this sentence. I can guess the meaning, but I find it quite misleading: apparently the idea is that it measures one parameter, that is true (but it is also true for Coulter counter and other instruments) and this is not the concept the authors wants to discuss here. "extinction length" is a property of a cloud of many particles impinged by light: please change.*

Line 72 (Line 72 – 74): Elaborated on sentence for clarity. Sentence now states, "However, it only provides a particle geometric size measurements in one dimension (i.e., particle length or width), and does not provide depth/height information for individual particles (Ruth et al., 2002; Simonsen et al., 2018)."

*line 82: "absorption intensity" is meaningless, please change. "light scatetring and absorption is proptional to the extinction cross section". More precisely, the extinction is the sum of scattering and absorption. please change.*

Line 82 (Line 81 - 84): Changed "absorption intensity" to "Particles are directed through the focal point of a light beam while sensors in the far field collect the total power removed from light scattering and absorption of the particle" We also elaborate on following sentence, "Because the sum of scattering and absorption by each particle is proportional to the extinction cross section, particle shape information can be identified for particles relatively quickly and efficiently (Potenza et al., 2015)."

*line 138-146: Why not comparing to a pure statistical estimate, based on the probability to have two particles in the Abakus scattering volume (if known)?*

Line 138 – 146: This technique was previously published and allows for a more accurate comparison of data cleaning to their dataset.  Furthermore, we do not know the Abakus scattering volume of the dataset and, rather than using a statistical estimate, we compare against our own data.

*line 170: Is Figure 1 actually S1?*

Line 170: Figure 1 or Figure S1 is not mentioned in that current paragraph.  We are unsure of what reviewer is referring to.

*line 182 and below: "Feret" with capital letter*

Line 182 and below: Replaced "ferret" with "Feret" throughout the entire document.

*line 203: "The flowCAM does not produce size estimates for size bins ....": please explain.*

Line 203 (Line 212 - 213): Reworded sentence, "The FlowCAM does not have the same size bin measurements as the Abakus and does not produce measurements for particle lengths between 2.1, 2.2, 2.4, and 2.7 µm."

*line 205: Figure S8 and S9 are mentioned before S7 (S6 is not mentioned). Figures should be presented in the order of appearance in the text. S8 and S9 appear before S7, while S6 is not mentioned in the text.*

Line 205: Reordered Supplementary Figure order so that the numbers are in order and added new Figure S6.

*line 210-211: If possible, this hypothesis should be better explained: it appears to be quite strong and requiring better support, especially in connection with the discussion about coincidences (line 138-146).*

Line 210 – 211 (216 – 222): We thank the reviewer for their comments and agree that the justification in this section was unclear as written. We have updated the text to provide further support and clarity. We add citations to Lines 141 – 145, "To check if the coarse particle concentration measured by the Abakus is composed of large particles rather than multiple small coagulated particles, we tested for coincidence following methods of Saey (1998), and used in Simonsen et al. (2018). Saey (1998) showed that during measurement of ice with high particle concentration (>240,000 particles mL$^{-1}$) two particles passing through the detector at the same time can be erroneously identified as one large particle (hereafter referred to as 'particle coincidence')." In Lines 216 – 223 we further reiterate that these established tests for coincide explain very little of the relationship between particles. Furthermore, because the Abakus was calibrated to CC samples (i.e., Koffman et al., 2014), we feel that the evidence supports our hypothesis in Lines 490 – 495.

*line 228: "binned by particle length under the assumption that length measurements of particles are equal between the Abakus and DPI". I would suggest the authors to slightly extend the discussion about this point: this assumption could generate a further bias, since Abakus is assumed here to give a well defined "length". Actually, it provides a much less defined "size" from the extinction cross section, that could be affected by shape and other particle features.*

Line 228: We add the following sentence recognizing these instrumental biases (Lines 255 – 257), "While we recognize that the Abakus may be recording extinction cross section rather than true particle length, we assume that both measurement techniques (Abakus and DPI) are representing accurate particle length since the Abakus was calibrated to CC measurements (Koffman et al., 2014)." However, we also add the following lines to Section 4.4 (Line 541 - 544), "We also note that the reduced discrepancy between the Abakus and CC data supports our hypothesis that DPI particle geometries can be matched with Abakus particle length (extinction cross section) measurements. If the Abakus geometric dimensions were extremely biased on millennial timescales, then we would expect the use of DPI to not reduce discrepancy between the two datasets."

*section 4.1, line 379: I expect that the drilling fluid form a kind of water emulsion. Do the authors have images obtained with these samples that can support this interpretation? Anyway I Agree to remove the corresponding data from the analysis.*

Line 379: We do not have any images of the drilling fluid within the samples that we took. The FlowCAM takes focused images of the particles rather than the background water.

*Line 391: Figure 6 is missing: is it actually Figure S6. Note that Figure S6 is not mentioned in the text.*

Line 391 (410): Figure 6 has been replaced with Figure 5 and Figure S6 is now mentioned in the text in Line 316.

*Finally, no mention is there of the unavoidable differences in the particle statistical orientation between DPI and Abakus. Works considering the bias in determining aspect ratios of non-spherical (mineral) particles observed at microscope could be a reference for a brief discussion of this point and it's consequences.*

Line 230 (240 – 247): Added sentence for clarity for description of assumptions regarding particle orientation and particle height dimension (See Major Comment 3). We also add (Lines 435 – 443) "Future studies should use this assumption with caution after measuring site specific particle metrics on the timescale in question. Mathaes et al. (2020) identified that measurement techniques can lead to greater variability of 5 µm spherical (aspect ratio = 1.0) or elongated particles (aspect ratio = 0.2). However, our aspect ratio measurements consistently cluster between values of 0.70 to 0.76 across a wide range of time periods and particle sizes (Figure S9). Not only are these aspect ratios outside the range of spherical and elongated particles (as defined by Mathaes et al. (2020)), but their consistency demonstrates the repeatability of our techniques. Despite our millennial-scale consistency in aspect ratios at the South Pole, Potenza et al. (2016) identified that during short decadal time periods (i.e., ~15 years), particle aspect ratio is variable, indicating that high-frequency changes in particle shape may exist despite the low-frequency stability"

Furthermore, in the Supplementary Material, we add to Figure S9 (Lines 73 – 75), "The error bars for two different aspect ratio measurements of 5 µm particles are shown for reference (Mathaes et al., 2020). It is likely that these error estimates are overly conservative as applied to our study because we used at 20x zoom factor (recommended by the FlowCam manual), whereas Mathaes et al. (2020) used a 10x zoom."

*As reported around line 420, clays typically dominate with an expected prevalence of oblate shapes. Is it possible to consider this piece of information to interpret data?*

Line 420 (Lines 475 - 478): Added in section addressing the dominant shape of oblate clay minerals and how our data compares to clay geometry. "While small clay particles (~2.5 µm) are predominantly oblate particle shapes, larger particles (~10 µm) became more variable in shape (Meland et al., 2012). The close correspondence between our $PSD_{Abakus}$ and $PSD_{CC}$ fine particles

supports their interpretation.  The disagreement between our $PSD_{Abakus}$ and $PSD_{CC}$ coarse particle further suggests that coarse particles are more variable in shape.

*In general, I suggest the authors to shift the equations before the corresponding discussion in the text.*

All equations are now before the corresponding discussion in the text.

*Figures should be presented in the order of appearance in the text. S8 and S9 appear before s7, while S6 is not mentioned in the text.*

All Supplementary Figures have been ordered correctly and appear in the text.

*S6 is not mentioned in the text.*

Figure S6 is now mentioned in Line 316.

*About Figures S6 and S7, the meaning is not clear. What are the distributions and the "Kernal density estimate"? A brief description could be beneficial for the readership.*

We add to Figure S6, "Kernel density estimates are calculated from a probability density function as an estimate for continuous random variables." For Figure S7 and S8 (previously S6 and S7), Figure S6 is highlighting the incorrect assumption that particle width is equal to particle length and Figure S8, shows the median width per size bin with 2σ variability in grey.  Black bars represent interpolated bin values.

---

## Author Comment (AC2)

We thank Reviewer 2 for their detailed and helpful comments.

*Major Comment 1: Dust in ice cores (and other paleoclimatic archives) usually produce aggregates where smaller particles stick together to form larger ones. These aggregates may have irregular and hollow shapes. Aggregates are usually broken apart using an ultrasound bath, but no such procedure was described in this manuscript. The authors mention large particles in relationship with drill fluid contamination, but some of those may also be aggregates. If possible, it would be good to get some SEM images from these sections. If not, I think aggregates and the potential effects on the presented measurements should be at least discussed. Note that this is different from the coincidence that the authors mention in line 139, which is a bit confuse. Coincidence is when for example two small particles are counted as on large one, as the authors mention in line 141, but when small particles coagulate (for example at the boundary of ice cristals) the we are talking about aggregates which is a different problem.*

This comment has been addressed by the addition of Lines 148 – 151 and 223 – 226. While some of the deeper core (samples from Heinrich Stadial 4 and 5) do show evidence of possible aggregation, we check the impact on our Abakus and CC samples.  We detail our methods of checking by adding the following (Line 149 – 152), "While we are aware of particle aggregation in deep sections of ice cores, we also access the impact of aggregation on PSD values.  Baccolo et al. (2018) identified particle aggregates in the deep sections of the Talos Dome core PSDs which were characterized by low fine particle concentration and a coarse (~4.4 µm) mode. We look for similar features in the deep South Pole Ice Core."

 In Lines 224 – 227, we also add the results of our particle aggregate test, "Furthermore, we find that the deeper core samples (Heinrich Stadial 4 and 5) PSD mode values compared to Heinrich Stadial 1 and LGM mode values are statistically similar (student t-test; p-value > 0.05). Therefore, while there may be particle aggregation, it is not having a significant effect on the deep South Pole samples."

*Major Comment 2: There is a bit of confusion on the binning and size distribution in the manuscript. In many places, the authors mention the bins of the Abakus, but in most places they seem to work with a coarse vs. fine fraction only. I think the manuscript could be made clearer about when the whole size distribution is used, and when only the coase vs. fine fraction. Related to this, the authors use the size fraction between 2 um and 6.4 um for their correction. However, it is not addressed in the text how these bins are to be found in the Abakus since there is no good way to attribute a specific voltage to a specific particle size, as the calibration with spherical latex particles does not work well. This should be clarified in the text, in particular the calibration of the Abakus particle sizes.*

Major Comment 2 Response: This comment has been addressed by the addition of Lines 366.

We add the following to describe the calibration of the Abakus in Lines 138 – 140, "The Abakus was calibrated using CC techniques and then successfully tested against latex spheres of 1, 2, 5, and 10 µm diameter to assess accuracy (Koffman et al., 2014). Therefore, our size range of 1.1 – 6.4 µm have been successfully used and are considered accurate."

For clarity of when we are referring to coarse vs fine particles or $PSD_{Abakus}$ and $PSD_{CC}$ we add the statement (Line 371 – 374), "Because of these differences in distribution and the non-linear offset between $PSD_{Abakus}$ and $PSD_{CC}$, we explore the differences between fine and coarse particles. For clarity, we refer to $PSD_{Abakus}$ and $PSD_{CC}$ when referring to the entire distribution of particles by size and fine and coarse particles when referencing generally trends based on distribution differences in aspect ratios."

*Major Comment 3: Chapter 3.1: This subchapter is unclear. It is not clear if this comparison in Fig S4 was made using the corrected Abakus PSD or the original. Also, Fig 3 appears to show different Abakus PSDs based on corrections, but these are not discussed at all in the text. One possibility would be to merge this chapter with chapter 3.3, for example.*

We add to the caption of Figure S4 detailing that the figure was calculated using spherical assumptions. We also add statements discussing changes relative the effect of shape in Figure 3. In Section 3.1 we add to the sentence in Line 320 - 322, "Furthermore, the distribution differences between $PSD_{Abakus}$ (under spherical shape assumption) and $PSD_{CC}$ are not temporally consistent (Figure 3 and S5), potentially indicating changes in particle shape over time.", by specifically stating the shape assumption and Figure 3.

We also add references to Figure 3, specifically stating under spherical assumption when discussing total offset in Lines 314. In Section 3.3, we add references to Figure 3 (Line 389), which are specifically discussing the effect of particle shape and reduction of our total offset parameter.

**Minor Comments**

*Line 40: The parenthesis with ~490 years per sample during the LGM is out of place. Put this information somewhere else.*

Line 40: Added "temporal resolution…" before ~490 years per sample during the LGM. We respectfully keep the placement of the 490 years to convey the average resolution of our samples.

*Lines 40-41: These are not period of rapid global climate reorganization (that would be the glacial termination), they just include some millennial scale variability, just as pretty much any other period except the Holocene.*

Line 40 – 41: Changed text from "…rapid global climate reorganization" …to "millennial scale climate variability".

*Line 42: number of samples is missing for HS4 (n=?)*

Line 42: Heinrich Stadial 4 and Heinrich Stadial 5 were grouped together in this analysis, where the n = 16 value represents the number of samples in both. The line has been changed so that now the number of samples in both Heinrich Stadial 4 and Heinrich Stadial 5 represents the number samples in each of those time periods for clarity.

*Line 70: I would say the Abakus provides theoretical millimeter-scale resolution, as the mixing of water in the tubes dilutes the original signal. Rasmussen et al. tried to deconvolute that back in the days (https://doi.org/10.1029/2004JD005717), but it was not really widely applicable.*

Line 70: Added that the Abakus provides theoretical millimeter-scale resolution (Line 71).

*Line 88: Why is the period 0-10 ka not included?*

Line 88: Due to sampling aliquots during melting, we had limited sample volume between 10 – 0 ka, which was used from trace element geochemistry analyses.

*Line 92: What's the accumulation rate during the glacial?*

Line 92 (Line 92 - 93): Added LGM accumulation rate (3.7; 27 – 18 ka) from Kahle et al. (2020).

*Figure 1: Maybe add EDC or Siple Dome CO2 to complete the record in panel B?*

Line 106: We add the WAIS Divide $CO_2$ record to Figure 1 since the South Pole Ice Core is stratigraphically tied to WAIS Divide (Epifanio et al., 2020; Winski et al., 2019).

*Line 175: Here there is some information missing. The CC is usually calibrated using commercially available spherical latex particles. How were the Abakus size bins calibrated and can they be directly compared to CC bins?*

Line 175 (Line 138 - 140): Added sentence describing Abakus calibration technique described in Koffman et al. (2014). "The Abakus was calibrated using CC techniques and then successfully tested against latex spheres of 1, 2, 5, and 10 µm diameter to assess accuracy (Koffman et al., 2014). Therefore, our size range of 1.1 – 6.4 µm have been successfully used and are considered accurate."

*Line 188-187: Similar comment here: The Abakus size ranges are notoriously difficult to calibrate. Are the FlowCAM aspect ratios really comparable to Abakus size ranges?*

Line 187 – 188: Per the FlowCAM user manual (Fluid Imaging Technologies, 2011; pg. 136); "The calibration constants are determined during the manufacturing of the instrument and should never be changed. Fluid Imaging Technologies, Inc. uses a proprietary optical calibration device to determine the calibration factor of each objective of the instrument. Added sentence in Line (now 195 - 196) stating "We use the default factory settings, which have a predetermined calibration factor, which converts pixels to micron measurements (Fluid Imaging Technologies, 2011)."

*Line 193: Figure S2 does not show particle counts by Abakus bin size.*

Line 193 (Line 202): Changed from "particle counts by Abakus bin size" to "particle counts by coarse particles (5.1 – 6.4 µm and total counts and Figure S3 for representative FlowCAM particle images from a single sample)."

*Line 210: Again, how were the Abakus size distributions calibrated? Could the offset come from an uncertain coarse:fine particle size threshold?*

Line 210: We refer to corrections of Line 175 (Line 138 – 140) and Line 187 – 188.

*Line 228: This is the crux of the problem. How can you be sure that length measurements of particles are equal between Abakus and DPI?*

Line 228: We refer to corrections of Line 175 (Line 138 – 140) and Line 187 – 188.

*Line 251: Again, how do you know which Abakus channels corresponds to the exact sizes 2 and 6.4 um?*

Line 251: We refer to corrections of Line 175 (Line 138 – 140) and Line 187 – 188.

*Lines 255-263: The mention of Eq.3 in the text could be a bit confusing since only eq. 3a and 3b are listed. Rephrase for clarity.*

Line 255 – 263 (285 - 295): Clarified in-text references to Equation 3 by adding a and b to respective statements.

*Line 286: In Figure S4 it says the slope is 1.81, not 2.3.*

Line 286 (Line 314): Corrected slope value from 1.81 to 2.32.

*Line 287-288: Could we please see the histograms in the SI?*

Line 287 – 288 (Line 317): Added figure to supporting information showing distributions of $PSD_{Abakus}$ and $PSD_{CC}$ (Figure S6).

*Line 295: Table 1 is first mentioned here, but appears only much further down in the document. I suggest moving it closer to the first mention.*

Line 295 (Line 339): Moved Table 1 from Line 394 to Line 339, so that it corresponds better to Section 3.1.

*Figure 4: The title says Particle aspect ratio as a function of time, but this is only true for panels b and d. Also, the y-axis line and title colors for Particle Concentrations and CO2 appear to correspond to the wrong curve in the plot. The last two sentences of the figure caption are redundant.*

Figure 4 (Line 356): We add to the opening sentence of Figure 4 caption that "Particle aspect ratio data as a function of aspect ratio (a and c) and time (b and d)." We also re-color the y-axis labels for clarity, and we remove the second to last sentence because of redundancy.

*Line 337: If you interpolated width values in bins 2.1, 2.2, 2.4, and 2.7, then why do these widths have multiple measurements and standard deviations in Figure S8?*

Line 337: Our initial attempt was to show the similarities between each of the three time periods under investigation. For clarity, we only produce figure now without subplots. The standard deviations represent the spread in the data from each of the time periods since multiple samples were taken during each period under investigation. We believe that the reviewer meant to say Figure S7 and have replaced the original figure with only one figure with 2σ error bars. If the reviewer was referring to Figure S8, there are no width measurements for bins 2.1, 2.2, 2.4, and 2.7µm but added the following sentences for clarity as to what error bars represent (Figure S8), "Error bars represent the variability of average width measurements throughout the record by particle size."

*Line 346: Closing parenthesis missing after 1c*

Line 346 (Line 387): Closed parenthesis after 1c.

*Line 443: This last phrase should be rephrased as particle numbers were not published in Ruth et al., 2008. Although the correlation between the two was very high, that does not imply the same absolute values.*

Line 443 (Line 504): Replaced the word value with relationships.

**References**

Baccolo, G., Cibin, G., Delmonte, B., Hampai, D., Marcelli, A., Di Stefano, E., Macis, S., and Maggi, V.: The Contribution of Synchrotron Light for the Characterization of Atmospheric Mineral Dust in Deep Ice Cores: Preliminary Results from the Talos Dome Ice Core (East Antarctica), Condensed Matter, 3, 25, 2018.
Epifanio, J. A., Brook, E. J., Buizert, C., Edwards, J. S., Sowers, T. A., Kahle, E. C., Severinghaus, J. P., Steig, E. J., Winski, D. A., Osterberg, E. C., Fudge, T. J., Aydin, M., Hood, E., Kalk, M., Kreutz, K. J., Ferris, D. G., and Kennedy, J. A.: The SP19 chronology for the South Pole Ice Core – Part 2: gas chronology, Δage, and smoothing of atmospheric records, Clim. Past, 16, 2431-2444, 10.5194/cp-16-2431-2020, 2020.
Fluid Imaging Technologies: FlowCAM Manual, 2011.
Kahle, E., Buizert, C., Conway, H., Epifanio, J., Fudge, T. J., and Jones, T. R.: Temperature, accumulation rate, and layer thinning from the South Pole ice core (SPC14) [dataset], doi: https://doi.org/10.15784/601396, 2020.
Koffman, B. G., Kreutz, K. J., Breton, D. J., Kane, E. J., and Winski, D. A.: Centennial-scale variability of the Southern Hemisphere westerly wind belt in the eastern Pacific over the past two millennia, Climate of the past, 10, 1125-1144, 10.5194/cp-10-1125-2014, 2014.

Mathaes, R., Manning, M. C., Winter, G., Engert, J., and Wilson, G. A.: Shape Characterization of Subvisible Particles Using Dynamic Imaging Analysis, Journal of pharmaceutical sciences, 109, 375-379, 10.1016/j.xphs.2019.08.023, 2020.

Potenza, M. A. C., Sanvito, T., and Pullia, A.: Measuring the complex field scattered by single submicron particles, AIP Advances, 5, 117222, 10.1063/1.4935927, 2015.

Potenza, M. A. C., Albani, S., Delmonte, B., Villa, S., Sanvito, T., Paroli, B., Pullia, A., Baccolo, G., Mahowald, N., and Maggi, V.: Shape and size constraints on dust optical properties from the Dome C ice core, Antarctica, Scientific Reports, 6, 9, 10.1038/srep28162, 2016.

Ruth, U., Wagenbach, D., Bigler, M., Steffensen, J. P., Rothlisberger, R., and Miller, H.: High-resolution micoparticle profiles at NorthGRIP, Greenland: case studies of the calcium-dust relationship, Annals of Glaciology, 35, 237-242, 2002.

Saey, P.: Diplomarbeit im Studiengang Physik, Fakultat fur Physik und Astronomie, Ruprecht-Karls-Universitat Heidelberg, 1998.

Simonsen, M. F., Cremonesi, L., Baccolo, G., Bosch, S., Delmonte, B., Erhardt, T., Kjær, H. A., Potenza, M., Svensson, A., and Vallelonga, P.: Particle shape accounts for instrumental discrepancy in ice core dust size distributions, Clim. Past Discuss., 19 (in press), 2018.

Winski, D. A., Fudge, T. J., Ferris, D. G., Osterberg, E. C., Fegyveresi, J. M., Cole-Dai, J., Thundercloud, Z., Cox, T. S., Kreutz, K. J., Ortman, N., Buizert, C., Epifanio, J., Brook, E. J., Beaudette, R., Severinghaus, J., Sowers, T., Steig, E. J., Kahle, E. C., Jones, T. R., Morris, V., Aydin, M., Nicewonger, M. R., Casey, K. A., Alley, R. B., Waddington, E. D., Iverson, N. A., Dunbar, N. W., Bay, R. C., Souney, J. M., Sigl, M., and McConnell, J. R.: The SP19 chronology for the South Pole Ice Core – Part 1: volcanic matching and annual layer counting, Clim. Past, 15, 1793-1808, 10.5194/cp-15-1793-2019, 2019.

---

## Author Response (AR2)

Dear Dr. Rousseau,

Thank you for the opportunity to revise our manuscript, "Non-spherical microparticle shape in Antarctica during the last glacial period affects dust volume-related metrics." Based on our conversation, I have edited the final document which includes the following changes.

Our responses are given below with reviewer comments in italics; author response in regular font, and revised text from the paper is provided in blue.

With Best Regards,
Aaron

Line 36: Changed 50 – 10 ka to 50 – 16 ka.

Line 40: Removed temporal resolution and added temporal resolution of each time period. Sentence now reads "The 41 discrete samples were collected during three periods of millennial scale climate variability: Heinrich Stadial 1 (18 – 16 ka, n = 6; ~250 years/sample), the LGM (27 – 18 ka, n = 19; ~460 years/sample), and during both Heinrich Stadial 4 (42 - 36 ka, n = 8; ~620 years/sample) and Heinrich Stadial 5 (50 – 46 ka, n = 8; ~440 years/sample)."

Line 363: Added space between $CO_2$ and (Bauska et al., 2021; salmon).

Figure S8: Added letters in figure and caption to match style of other figures.

Figure S9: Edited letter's parentheses used in the figures matched other parentheses. Caption names were also changed to reflect particle shape instead of shorthand form (i.e., prolate vs El. 1). The figure caption has also been correspondingly changed.

Figure S11: Edited letter's parentheses in figure to match parentheses style of other figures.

Figure S12: Edited letter's parentheses in figure to match parentheses style of other figures.

Supplementary Material: Added Refences label before first reference.

Supplementary References: Removed duplicate references